# Nonlinear Guided Wave Tomography for Detection and Evaluation of Early-Life Material Degradation in Plates

**DOI:** 10.3390/s21165498

**Published:** 2021-08-16

**Authors:** Chengwei Zhao, Sunia Tanweer, Jian Li, Min Lin, Xiang Zhang, Yang Liu

**Affiliations:** 1State Key Laboratory of Precision Measuring Technology and Instruments, Tianjin University, 92 Weijin Road, Tianjin 300072, China; zhaochengwei_jlu@163.com (C.Z.); tjupipe@tju.edu.cn (J.L.); 2Department of Mechanical Engineering, School of Mechanical and Manufacturing Engineering, National University of Sciences and Technology, Islamabad 44000, Pakistan; stanweer.me17bsmme@student.nust.edu.pk; 3Department of Mechanical Engineering, University of Wyoming, 1000 E. University Ave, Laramie, WY 82071, USA; mlin4@uwyo.edu (M.L.); xiang.zhang@uwyo.edu (X.Z.)

**Keywords:** nonlinear ultrasonic guided waves, early-life material degradation, Lamb waves, third order elastic constants, Murnaghan’s model

## Abstract

In this paper, the possibility of using nonlinear ultrasonic guided waves for early-life material degradation in metal plates is investigated through both computational modeling and study. The analysis of the second harmonics of Lamb waves in a free boundary aluminum plate, and the internal resonance conditions between the Lamb wave primary modes and the second harmonics are investigated. Subsequently, Murnaghan’s hyperelastic model is implemented in a finite element (FE) analysis to study the response of aluminum plates subjected to a 60 kHz Hanning-windowed tone burst. Different stages of material degradation are reflected as the changes in the third order elastic constants (TOECs) of the Murnaghan’s model. The reconstructed degradations match the actual ones well across various degrees of degradation. The effects of several relevant factors on the accuracy of reconstructions are also discussed.

## 1. Introduction

Nonlinear ultrasonic guided waves combine the advantages of nonlinear ultrasound and ultrasonic guided waves and have attracted growing interest for nondestructive evaluation (NDE) in recent decades [1]. The great sensitivity of nonlinear ultrasonic guided waves in detecting early-life material degradation owing to fatigue, creep, thermal aging, embrittlement, stress corrosion cracking, etc., makes them advantageous over the linear ultrasonic guided waves when detecting early material degradation or small-sized damage, i.e., smaller than the wave length of the ultrasound, and is of interest [2].

Since the 1960s, Goldberg [3] and Rollins [4] have theoretically analyzed the propagation of elastic waves in a uniform and continuous solid media. Later, Deng first investigated the cumulative second harmonics of horizontal shear (SH) waves and Lamb waves, and further studied the generation process and symmetric relation of cumulative second harmonics with appropriate boundary and initial conditions of excitation [5,6]. Following these findings, de Lima and Hamilton evaluated the power transferred from the primary waves to the modes in the expansion of the secondary waves, and explained harmonic generation in homogeneous, isotropic, stress-free elastic waveguides with arbitrary constant cross-sectional area [7]. Subsequently, Srivastava and Lanza di Scalea [8], Müller et al. [9], Chillara and Lissenden [10], and Liu et al. [1,11] investigated higher-order harmonic generation in weakly nonlinear elastic plates and developed the theories of nonlinear ultrasonic guided waves.

In the area of the coupling mechanism of the microstructure, Cantrell and Yost et al. proposed the dependence of a model of acoustic harmonic generation in polycrystalline solids on the coherency strains, which proved the potential useful means of nonlinear ultrasonic guided waves in optimizing the treatment time for NDE [12]. Deng and Pei proved that the effect of second-harmonic generation by Lamb wave propagation is sensitive to the accumulation of fatigue cracks in solid plates [13]. Moreover, Kim et al. [14], Cash and Cai [15], Xiang et al. [16], Li et al. [17] and Lissenden et al. [18] conducted extensive studies on nonlinear guided waves in experimental research.

In order to reconstruct the position, size and degree of defects, linear ultrasonic guided waves are often combined with tomography, e.g., ray tomography [19], diffraction tomography [20] and full wave inversion [21]. However, a literature review indicates a paucity in the combination of nonlinear ultrasonic guided waves and tomography through a numerical establishment of a hyperelastic model [1,2,11,18].

Murnaghan’s model holds the ability to describe acoustic-elasticity and dynamic nonlinear elasticity in hyperelastic materials [22]. The theory of elasticity shows that with linear kinematic equations, the physical nonlinearity is formally associated with the nonlinear relationship between the stress tensor and strain tensors [23]. Therefore, Murnaghan’s model acts as a weakly nonlinear model, as opposed to the linear (Hookean) model [24,25], to define the various degrees of degradation. On the other hand, it was demonstrated that the third-order elastic constants (TOECs) in Murnaghan’s model are sensitive to fatigue damage [26]. In the early stage of degradation, the TOECs of the material will increase with the degree of fatigue [27]. Therefore, regions with different degrees of material early-life degradation in a structure can be represented by varying the TOECs of the intact material in Murngahan’s model.

In this work, the use of nonlinear guided wave tomography for detection and evaluation of early-life material degradation in metal plates is studied. This manuscript presents the following novel contributions: (1) Nonlinear ultrasonic guided waves in a Murnaghan material are combined with tomography. (2) A generalized nonlinear acoustic modeling framework has been established via the implementation of hyperelastic constitutive relations into Abaqus VUMAT subroutine. (3) Numerical results validated that nonlinear guided wave tomography is capable of characterizing the severity of early-life material degradation. (4) A guided wave signal processing platform has been developed for tomographic imaging with abundant acoustic features, including nonlinear harmonics, attenuation and signal difference coefficient (SDC), etc.

The remainder of the paper is organized as follows: Section 2 starts with an overview of the second harmonics of Lamb waves in a free boundary plate, then analyzes the internal resonance conditions between the Lamb wave primary modes and the secondary modes. The implementation of Murnaghan’s model as an Abaqus user material subroutine VUMAT is established in Section 3, which provides a connection between linear elasticity and weak nonlinearity. Section 4 covers the nonlinear model construction of plates in Abaqus for the FE simulations. Section 5 devises the trigonometric operations for the centripetal receiving, and the nonlinear guided wave (NGW) method for tomography to reconstruct the degradation in the plate. The reconstruction results are discussed and the effects on the reconstruction accuracy from multiple relevant factors are discussed. Finally, Section 6 provides concluding remarks and suggests future works.

## 2. Theoretical Preliminaries

A comprehensive review of the physical principles of nonlinear ultrasonic guided waves is provided by Lissenden [2]. In this section, a brief overview of the properties of cumulative Rayleigh–Lamb (RL) secondary modes is presented for building a full framework [1,2,7].

In order to formulate the field equations of nonlinear guided waves, the Lagrangian description is used, as shown in Figure 1. An elastic potential function, *W*, of Murnaghan’s model can be expanded as [22,28]:
(1)$$W=\frac{\lambda }{2}{\left[\mathrm{tr}\left(E\right)\right]}^{2}+\mu \mathrm{tr}\left(E^{2}\right)+\frac{\nu_{1}}{6}{\left[\mathrm{tr}\left(E\right)\right]}^{3}+\nu_{2}\mathrm{tr}\left(E\right)\mathrm{tr}\left(E^{2}\right)+\frac{4\nu_{3}}{3}\mathrm{tr}\left(E^{3}\right)+O\left(E^{4}\right)$$
where tr(.) denotes the trace of the bracketed tensor, λ and μ are Lamé constants, $\nu_{i}\left(i=1\;\mathrm{to}\;3\right)$ are Murnaghan’s TOECs, which are directly related to Landau–Lifshitz’s TOECs [22,25], and $E=\left[H+H^{T}+H^{T}H\right]/2$ are the invariants of the Lagrangian strain tensor which, related to the displacement vector H=∇u. The second Piola–Kirchhoff stress tensor, ***T***_*PK*2_, can be obtained from *W*:(2)$$\begin{array}{c} T_{PK2}=\frac{\partial W}{\partial E}=\lambda \mathrm{tr}\left(E\right)I+2\mu E+\frac{\nu_{1}}{2}{\left[\mathrm{tr}\left(E\right)\right]}^{2}I+\nu_{2}\mathrm{tr}\left(E^{2}\right)I\\ +2\nu_{2}\mathrm{tr}\left(E\right)E+4\nu_{3}\mathrm{tr}\left(E^{2}\right)+O\left(E^{3}\right) \end{array}$$
where ***I*** is the identity tensor. The equation of motion is usually given by the first Piola–Kirchhoff stress tensor, ***T***_*PK*1_, which is related to ***T***_*PK*2_ through the deformation gradient F=I+H [29]. In addition, ***T***_*PK*1_ can be decoupled into a linear component, $T_{PK1}^{L}$, and a nonlinear component, $T_{PK1}^{NL}$, as:(3)$$T_{PK1}{=T}_{PK1}^{L}{+T}_{PK1}^{NL}$$
(4)$$T_{PK1}^{L}=\lambda \mathrm{tr}\left(H\right)I+\mu \left(H+H^{T}\right)$$
(5)$$\begin{array}{c} T_{PK1}^{NL}=\frac{\nu_{1}}{8}H^{T}H^{T}+\left(\mu +\frac{\nu_{1}}{8}\right)\left(H^{2}+H^{T}H+{HH}^{T}\right)+\nu_{2}\mathrm{tr}\left(H\right)H^{T}+\frac{\nu_{2}}{2}\mathrm{tr}\left(H^{2}+H^{T}H\right)I\\ +\left(\lambda +\nu_{2}\right)\mathrm{tr}\left(H\right)H+\left(\frac{\lambda }{2}\mathrm{tr}\left[H^{T}H\right]+4\nu_{3}{\left(\mathrm{tr}\left(H\right)\right)}^{2}\right)I+O\left(H^{3}\right) \end{array}$$

The nonlinear wave field can be decomposed into a primary component, $u^{\left(1\right)}$, and a secondary component, $u^{\left(2\right)}$. Auld’s reciprocity relation can be used to prove the solving process of the primary wave field [30]. According to the normal mode expansion, the generated secondary component can be expressed as [7]:(6)$$u^{\left(2\right)}\left(X_{1},X_{3},t\right)=\frac{1}{2}\sum_{m=1}^{\infty }A_{m}\left(X_{1}\right)u_{m}\left(X_{3}\right)e^{-i2\omega t}$$
where *ω* is the angular frequency and $A_{m}\left(X_{1}\right)$ represents the modal amplitude, which can be obtained by the reciprocity theorem as:(7)$$A_{m}=\frac{f_{n}^{surf}+f_{n}^{vol}}{4P_{mn}}\{\begin{array}{l} \frac{i}{k_{n}^{*}-2k}\left(e^{i2kX_{1}}-e^{ik_{n}^{*}X_{1}}\right),\;\mathrm{for}\;k_{n}^{*}\neq 2k\\ X_{1}e^{i2kX_{1}},\;\mathrm{for}\;k_{n}^{*}=2k \end{array}$$
where $f_{n}^{surf}$ and $f_{n}^{vol}$ are the nonlinear driving forces which transfer the power flux from the primary mode to the secondary mode through the surface and volume, respectively. $P_{mn}$ is the complex power flux in the wave propagation direction, which is related to the modal velocity, $\nu_{m}$, and stress, $T_{m}$:(8)$$f_{n}^{surf}=-\frac{1}{2}\left(\nu_{n}^{*}\cdot T_{PK1}^{NL\left(1,1\right)}\right)\cdot n_{X_{3}}|{}_{-h}^{h}$$
(9)$$f_{n}^{vol}=\frac{1}{2}\int_{-h}^{h}\nu_{n}^{*}\cdot \left(\nabla \cdot T_{PK1}^{NL\left(1,1\right)}\right)\;dX_{3}$$
(10)$$P_{mn}=-\frac{1}{4}\int_{-h}^{h}\left(\frac{\nu_{n}^{*}}{2}\cdot \frac{T_{m}}{2}+\frac{\nu_{m}}{2}\cdot \frac{T_{n}^{*}}{2}\right)\;\cdot n_{X_{1}}dX_{3}$$
where $T_{PK1}^{NL\left(1,1\right)}$ corresponds to the nonlinear stress.

As indicated by Equation (7), in order to make the amplitude of a cumulative second harmonic to increase linearly with the propagation distance, the following conditions should be considered: (1) synchronism (phase matching), $k_{n}^{*}=2k$; (2) nonzero power flux between the primary and secondary modes, i.e., $f_{n}^{surf}+f_{n}^{vol}\neq 0$ [1,31,32]. Moreover, it has been proven that the nonzero power flux only occurs on symmetric RL second harmonic modes, and only the phase matching of symmetric RL second modes is under consideration [5,7,11,12].

Therefore, according to the material properties and the plate’s thickness, this work will bring out phase matching points from the dispersion curves plotted and choose the mode pair of specific frequency among the points to actuate the ideal nonlinear guided wave.

## 3. Implementation of Hyperelastic Material Model

In order to implement Murnaghan’s model, the elastic potential function, *W*, needs to be expressed as a function of invariants to eliminate the volume change [33]. The invariants, Green deformation tensors and stretch tensors can be decomposed into their dilatational and distortional parts, leading Equation (1) to be expressed as [34]:(11)$$U\left({\bar{I}}_{1},{\bar{I}}_{2},J\right)=a_{1}J^{2/3}{\bar{I}}_{1}+a_{2}J^{4/3}\;{\bar{I}}_{1}{}^{2}+a_{3}J^{4/3}{\bar{I}}_{2}+a_{4}J^{2}{\bar{I}}_{1}{\bar{I}}_{2}+a_{5}J^{2}{\bar{I}}_{1}{}^{3}+a_{6}\left(J^{2}-1\right)+a_{7}$$
where $\bar{C}=J^{-2/3}F^{T}F={\bar{F}}^{T}\bar{F}$ is the left Cauchy–Green deformation tensor with the first and second distortional invariants ${\bar{I}}_{1}$ and ${\bar{I}}_{2}$, i.e., ${\bar{I}}_{1}=\mathrm{tr}\bar{C}$, ${\bar{I}}_{2}=\frac{1}{2}\left[{\left(\mathrm{tr}\bar{C}\right)}^{2}-\left(\mathrm{tr}{\bar{C}}^{2}\right)\right]$. $J=T_{PK1}F^{T}/\sigma$ is the total volume ratio for the spatial description of equilibrium requiring the Cauchy stress tensor, and $a_{i}\left(i=1\;\mathrm{to}\;7\right)$ are material constants, which can be expressed as a linear combination of Lamé constants and Murnaghan’s TOECs [34].

Recall the relation of Cauchy true stress with elastic potential function and left Cauchy–Green deformation tensor [35], which as shown in the Appendix B leads to:(12)$$\begin{array}{c} \sigma =2\left(\frac{a_{1}}{J^{1/3}}+2a_{2}J^{1/3}{\bar{I}}_{1}+a_{4}J{\bar{I}}_{2}+3a_{5}J{\bar{I}}_{1}{}^{2}+a_{3}J^{1/3}{\bar{I}}_{1}+a_{4}J{\bar{I}}_{1}{}^{2}\right)\bar{B}\\ -2\left(a_{3}J^{1/3}+a_{4}J{\bar{I}}_{1}\right)\bar{B}\cdot \bar{B}+\left(2a_{6}J\right)I \end{array}$$
where $\bar{B}=\bar{F}{\bar{F}}^{T}$ is the left Cauchy–Green deformation tensor.

Implementation of Murnaghan’s model in Abaqus as a VUMAT requires a frame-independent definition of stress. This can be achieved by writing the distortional component of left Cauchy–Green deformation tensor in terms of distortional component of right stretch tensor, $\bar{U}$, as shown in the Appendix B:(13)$$\begin{array}{c} \sigma =R[\left(\frac{2a_{1}}{J^{1/3}}+4a_{2}J^{1/3}{\bar{I}}_{1}+2a_{4}J{\bar{I}}_{2}+6a_{5}J{\bar{I}}_{1}{}^{2}+2a_{3}J^{1/3}{\bar{I}}_{1}+2a_{4}J{\bar{I}}_{1}{}^{2}\right){\bar{U}}^{2}\\ -\left(2a_{3}J^{1/3}+2a_{4}J{\bar{I}}_{1}\right){\bar{U}}^{4}+\left(2a_{6}J\right)I]R^{T} \end{array}$$
where the factor in the bracket can be recognized as the frame-independent, objective counterpart of the Cauchy stress tensor and ***R*** is the rotation tensor from polar decomposition of the deformation gradient. This numerical implementation of Murnaghan’s model is applicable to isotropic hyperelastic materials. More details of the implementation are provided in the Appendix B.

## 4. Nonlinear Model Construction in Plates

### 4.1. Primary Mode Selection of Lamb Waves

It has been discussed that symmetric Lamb waves in plates generate cumulative second harmonics under some special cases in Section 2. In fact, each internal resonance point is valid in a region that is dependent on the frequency bandwidth associated with the toneburst excitation and phase velocity bandwidth associated with the finite sensor size [2,31,36]. In this work, the nonlinear models were used to simulate by using aluminum plates with d=2h, and the material parameters representative of aluminum are listed in Appendix A [22,37]. The phase velocity dispersion curves of the RL waves of the aluminum plate are plotted in Figure 2. The *fd* product for the primary modes, and 2*fd* for the second harmonics have been provided. The primary modes are plotted with solid lines and identified by upper case letters (e.g., A0, A1, S0, S1), and the secondary modes are plotted with dash lines and identified by lower case letters (e.g., s0, s1, s2, s3), where A and S(s) represent the antisymmetric mode and the symmetric mode, respectively.

A number of internal resonance points satisfying both synchronism and nonzero power flux are marked in Figure 2, which are represented as primary–secondary mode pairs. Although the S1–s2 mode pair is commonly used in a plate as a nonlinear Lamb wave measurement [1,2], it is difficult to preferentially actuate the S1 mode at the point due to the proximity of other modes (especially A1 mode) and the wide variety of modes. However, the S0–s0 mode pair has the ability to avoid multi-mode interference. On the other hand, the S0 primary mode has a strong excitation and minimal dispersion at the low frequency region, and the s0 secondary mode has a strong receptivity [32]. Therefore, the S0 mode Lamb wave at the internal resonance point of fd=0.6 MHz·mm was chosen as the primary mode. The phase velocity *c_p_* dispersion curves of the primary modes in Figure 2 are converted into the group velocity *c_g_* dispersion curves and are plotted in Figure 3. At fd=0.6 MHz·mm, only A0 and S0 modes exist as the primary modes in the aluminum plate, whose *c_g_* differ greatly. This is conducive to actuation of pure S0 mode, identifying and extracting the S0 mode among the time domain signals.

However, the in-plane displacement is dominant at fd=0.6 MHz·mm, and the power flux of the S0–s0 mode pair is relatively weak. It needs to involve the in-plane displacement component of the wave actuation and reception, rather than the out-of-plane displacement component at the surface of the plate. For signal collection, a beam sensor has a certain angle, and each sensor in the array is set for centripetal actuation and in-plane reception.

### 4.2. Design of Finite Element Simulations

Reconstruction algorithm for probabilistic inspection of damage (RAPID) is an algorithm based on correlation analysis. By comparing the reference signals with the defect detection signals, the signal differences can allow defect identification [38]. In this work, RAPID based on ray tomography was used for reconstruction. Two aluminum plate models with and without degradation were established, whose parameters were set the same except for the degraded regions, following the findings that degradation can be accounted for by using higher TOECs in the Murnaghan model [26,27]. The aluminum plate model has a size of 1200 mm×1200 mm×10 mm, which was spatially discretized into a structured mesh using 1 mm cubic C3D8R elements. Thus, the total number of elements was $1200\times 1200\times 10=1.44\times 10^{7}$ and the total number of nodes was 1201×1201×11=15866411. The actual plate was the central 1000 mm×1000 mm×10 mm segment, while the remaining boundary layer was set for absorbing the Lamb waves to avoid reflection. Coordinate (0, 0) is the center of the plate which was set as the origin. The circular sensor array was installed on the nodes around the degradations, and the separation distance *D* between the sensors, that deployed along a circular aperture of radius, *r*, has to satisfy an ideal condition [39]:(14)$$D=\sqrt{2r^{2}-2r^{2}\cos\left(\frac{2\pi }{SN}\right)}<\frac{\lambda_{p}}{2}$$
where *SN* is the number of sensors, and $\lambda_{p}$ is the wavelength of the *c_p_* of the primary mode. At the internal resonance point of fd=0.6 MHz·mm, the primary mode of S0 has $c_{p}=5.366\;\mathrm{km}/s$ and $\lambda_{p}=89.43\;\mathrm{mm}$.

The resolutions are hardly improved again as *D* shrinks when Equation (14) has been satisfied. In order to save the computational cost, the parameters of the sensor array are set as: r=400 mm, SN=64. As shown in Figure 4, three circular degradations with diameters $d_{N1}=d_{N2}=d_{N3}=60\;\mathrm{mm}$, and depths $h_{N1}=h_{N2}=h_{N3}=5\;\mathrm{mm}$ are defined on some nodes on the aluminum plate, whose coordinates of the center are (−200, −200), (100, −100) and (0, 200), respectively. The TOECs ($\nu_{1}$, $\nu_{2}$, and $\nu_{3}$) of the degraded regions are modelled as 3×, 5×, and 10× of that of the intact material to indicate different degrees of degradation [26,27].

A 20-cycle Hanning-modulated tone burst excitation with a central frequency of 60 kHz was sent to the sensors, the displacement output produced by which was taken as the vibration signal of the piezoelectric sensors. For every sensor, it receives both the signal actuated by itself and the signal actuated by the opposite sensor, as shown in Figure 5. The time for the sensor to receive the positive peak of its own wave packet is $t_{1}=0.2135\times 10^{-3}\;s$. For a propagation distance of 800 mm, the time of the positive peak of the first wave packet actuated by the opposite sensor is $t_{1}+t_{2}=0.365\times 10^{-3}\;s$. The *c_g_* of the first wave packet is calculated to be 5.281 km/s, which corresponds to the point of S0 mode at fd=0.6 MHz·mm in Figure 3. It is indicated that the first wave packet is attributed to the S0 Lamb wave primary mode in Figure 5c. Similarly, the *c_g_* of the second wave packet is calculated to be 2.963 km/s, corresponding to the primary mode of A0. The frequency spectra for the received signals after fast Fourier transform (FFT) are also shown in Figure 5. After a propagation distance of 800 mm, the accumulated second harmonics are clearly observed in the signals received by the opposite sensor. Moreover, RL wave motion is noticed at the third and higher-order harmonic frequencies when the second harmonics are generated. Compared with the primary modes, the power flux of nonlinear harmonics is weak. However, the second harmonics dominate the nonlinear harmonics, and it is expected that reconstructions with second harmonics will have a higher resolution.

## 5. Tomography

### 5.1. Signal Processing of In-Plane Displacements

In a sensor array, every two sensors can be formed as an actuation–reception pair, and all actuation–reception pairs form a network over the detected area. In the *X*_1_–*X*_2_ plane, the sensor located in the positive direction of *X*_1_ was labelled as sensor 1, and increased the label number by 1 in the counterclockwise direction, as shown in Figure 6.

The displacement signals *S*_1_ and *S*_2_ at each sensor are collected from Abaqus in *X*_1_ and *X*_2_ directions, respectively. Therefore, it is necessary to calculate the angle *θ* between the vector of each actuation–reception pair and *X*_1_ direction, as shown in Figure 6. Equation (15) is then used for the operation which combines *S*_1_ and *S*_2_ into the actuation–reception direction signal, *S*.
(15)$$S=\pm S_{1}\times \left|\cos\theta \right|\pm S_{2}\times \left|\sin\theta \right|$$
where the positive and negative signs relate to the direction of the vector for each actuation–reception pair. For example, when the vector of an actuation–reception pair (*i–j* pair) is towards the Quadrant IV of the reference coordinate system, as shown in Figure 6, the result is:(16)$$\begin{array}{c} S=S_{1}\times \left|\cos\theta \right|-S_{2}\times \left|\sin\theta \right|\\ =S_{1}\times \left|\cos\left[\pi -\frac{\pi -\left(\beta -\alpha \right)}{2}-\beta \right]\right|\\ -S_{2}\times \left|\sin\left[\pi -\frac{\pi -\left(\beta -\alpha \right)}{2}-\beta \right]\right| \end{array}$$
where β=2π(i−1)/64 and α=2π(j−1)/64. Note there are some special cases, such as when the vector is parallel to the *X*_1_ or *X*_2_: one term in Equation (15) vanishes, and when i=j (α=β), Equation (15) becomes:(17)$$S=S_{1}\times \cos\beta +S_{2}\times \sin\beta$$

### 5.2. Reconstruction Using Nonlinear Harmonic Signals

Due to the inherent dispersion and multi-mode properties of guided waves, a time-frequency representation of the wave field has been created by using the short time Fourier transform (STFT), and a Hanning window was added to reduce spectrum leakage [1]. Then, the time intervals of the S0 mode wave packets and the frequency intervals of the second harmonics were intercepted, and the matrices representing the power flux of the second harmonics were selected from the STFT. By integrating the matrices, the SDC, *C_ij_*, of each *i–j* pair were calculated as:(18)$$C_{ij}=\left|1-\frac{Y_{ij}}{Z_{ij}}\right|$$
where *Y_ij_* is the matrix integral of the *i–j* pair in the model without degradation, and *Z_ij_* is the matrix integral of the *i–j* pair in the model with degradations. The value of *C_ij_* is proportional to the probability of degradation between the actuation–reception pair.

In this work, all the SDCs were transformed into the probability distribution of degradations by the elliptic algorithm. In order to determine the location, it was assumed that a degradation occurrence at a certain point could be estimated based on the SDC of a different *i–j* pair as a result of this degradation and its position relative to the *i–j* pair. In the presence of degradation, the most significant signal change was in the direct wave path, and the signal change effect decreased if the degradation is away from this path of the *i–j* pair [40]. Therefore, the probability distribution was expressed as a linear accumulation of all the signal change effects of each *i–j* pair. Each *i–j* pair had a spatial distribution. As shown in Figure 7, *i* and *j* are on the two vertices of the ellipse, and the value of SDC set at the line connecting *i* and *j* decreases from the middle to both sides. In the array network, the estimation of distribution probability, $P(X_{1},X_{2})$ at position $\left(X_{1},X_{2}\right)$ can be written as:
(19)$$P(X_{1},X_{2})=\sum_{i=1}^{SN-1}\sum_{j=i+1}^{SN}P_{ij}\left(X_{1},X_{2}\right)=\sum_{i=1}^{SN-1}\sum_{j=i+1}^{SN}C_{ij}\left[\frac{b-A_{ij}\left(X_{1},X_{2}\right)}{b-1}\right]$$
where $P_{ij}\left(X_{1},X_{2}\right)$ represents probability estimation of the degradation distribution from the *i–j* pair, and $\left[\left(b-A_{ij}(X_{1},X_{2})\right)/\left(b-1\right)\right]$ is the non-negative linear decreasing spatial distribution function of the *i–j* pair, as shown in Figure 7, the outline of which is a set of ellipses. Among them:(20)$$A_{ij}\left(X_{1},X_{2}\right)=\{\begin{array}{l} B_{ij}\left(X_{1},X_{2}\right),\;\mathrm{for}\;B_{ij}\left(X_{1},X_{2}\right)<b\\ b,\;\mathrm{for}\;B_{ij}\left(X_{1},X_{2}\right)\geq b \end{array}$$
where $B_{ij}(X_{1},X_{2})$ represents the ratio of the sum of distance between point ($X_{1},X_{2}$) to *i* and *j* to the distance between *i* and *j*, and *b* is a scaling parameter that controls the size of the effective elliptical distribution area. Usually, *b* is chosen to be around 1.05 [40]. The resolution of reconstruction can be improved by appropriately adjusting the value of *b*.

### 5.3. Effect of Degradation Degrees

In the early stage, fatigue is an important factor in degradation [2]. The early detection of this degradation contributes to early maintenance, leading to an increase in the service life. Therefore, we tested the capability of our modeling in detecting different levels of degradation.

As described in Section 4.2, degradation degrees were represented as multiples of $\nu_{1}$, $\nu_{2}$, and $\nu_{3}$. The probability distribution of the aluminum plate model by using the NGW method is shown in Figure 8, and the signal difference is expressed as a heat map. It can be observed that there are three circular degradations in the reconstruction, the positions of which approximately conform to the FE model with slight deviation. A possible reason for the position deviation is that there is a superposition of stress changes among these three degradations. It is also noticed that the position deviation is smaller for the degradation with higher degree, which suggests that detecting a smaller and lower degree of degradation is harder in general. However, relatively accurate prediction indicates that second harmonic tomography can reconstruct multiple degradations with various degrees in plates.

Compared with degradation No. 2 and No. 3, No. 1 describes an early degradation stage; as shown in Figure 8b, due to the change in stress caused by variation in TOECs, the difference of power flux exists around the degradations. Particularly, there is a superposition between No. 2 and No. 3, and consequently the signal difference is over the peak value of the degradation No. 1, which explains why the resolution of No. 1 is lower. However, the location of No. 1 can still be detected reasonably well, which reveals that the second harmonic tomography can be applied to detect much earlier stages of degradation.

The peak values of total SDC of the three degradations were obtained as we can see from Figure 8a and then plotted with TOECs in Figure 9a. It can be observed that different degradation degrees in a wide range have a linear relation with the value of SDC with a relatively high slope. This indicates that the sensitivity of second harmonic detection degradation is positively correlated with TOECs. In other words, the SDCs can be used to predict the variation of TOECs. Therefore, the NGW method has the potential capacity to simultaneously realize the qualitative detection of material degradation and the quantitative detection of the degradation degree. In addition, the above results show that a generalized nonlinear acoustic modeling framework has been established via the implementation of hyperelastic constitutive relations into Abaqus VUMAT subroutine, in which the solutions from linear elasticity can accurately express the stress change of Murnaghan’s materials.

### 5.4. Effect of Signal Processing Methods

A guided wave signal processing platform had been developed for tomography with abundant acoustic features, including nonlinear harmonics, attenuation and SDC, etc. [40]. This section will compare the results reconstructed by the attenuation, maximum peak (MP), covariance and NGW methods, whose peak values of total SDC with TOECS are all plotted in Figure 9a.

The principal characteristic of the attenuation method is to select the peak prominences in the time domain signals of each *i–j* pair and use them to calculate *C_ij_*. The probability distribution by using the attenuation method is shown in Figure 10a, which shows the accurately located position of the three degradations, but with shape distortions and imaging artifacts. In addition, the peak values of total SDC of the attenuation method do not have a linear relationship with TOECs, as shown in Figure 9a, which shows that it is not capable of characterizing the severity of early-life material degradation. Unlike the attenuation method, the MP method uses the peak values in the time domain signals of each *i–j* pair to calculate *C_ij_*. The probability distribution by using the MP method is shown in Figure 10b. Contrary to expectations, there are three locations surrounded by higher SDCs, the positions of which approximately conform to the degradations in the FE model. As a result, it is easy to mislead the reconstruction, so this method is inappropriate for detection of material degradation. The covariance method has the most extensive application in linear ultrasonic guided wave tomography, which directly calculates the signal covariance between each *i–j* pair in the two models. The probability distribution by using the covariance method is shown in Figure 10c. It is clearly indicated that degradations cannot be detected by the covariance method under different $\nu_{1}$, $\nu_{2}$, and $\nu_{3}$ conditions.

Compared with the three methods mentioned above, the most significant feature of the NGW method is the ability to extract second harmonics separately. The amplified version of Figure 5d for the primary mode and two nonlinear harmonic modes is shown in Figure 9b. The peak value of the primary mode is $12.98\times 10^{-9}$, and the power flux is $45.03\times 10^{-9}$ by calculating the peak area. The peak value of the secondary mode is $0.28\times 10^{-9}$, and the power flux is $1.826\times 10^{-9}$. The power flux intensity of the secondary mode is different from that of the primary mode, which is two orders of magnitude lower, resulting in the coverage of the signal differences detected by the second harmonics.

Different from linear guided waves, the existence and characteristics of defects in materials detected by nonlinear guided waves are related to an acoustic signal whose frequency differs from that of the actuation signal [41]. The attenuation, MP and covariance methods are suitable for the detection of macroscopic defects by using the primary modes, whereas the NGW method is suitable for analyzing the power flux of the second harmonics separately, and is able to detect early degradations. Hence, second harmonics are sensitive to the change of the constitutive relation of the microstructure, that is, they are sensitive to the early-life material degradation.

### 5.5. Effect of Harmonic Orders

While the second harmonics were generated, RL wave motion also existed at the third harmonic frequencies, as shown in Figure 9b. It can be observed that the third harmonics also have obvious cumulative effect of power flux in the frequency spectra of the signals received by the opposite side sensor. Therefore, a numerical study was carried out to study the effect of harmonic orders.

The third harmonics were used with the NGW method, and the probability distribution of aluminum plate models with early-life material degradations is shown in Figure 11. It can be observed that there are three circular degradations in the reconstruction, which are basically the same as those in Figure 8. As shown in Figure 9b, the peak value of the cubic mode is $0.11\times 10^{-9}$, and the power flux is $0.81\times 10^{-9}$, which does not reach 1/2 of the secondary mode. However, compared with the second harmonic, the overall signals differ little in the probability distribution of the third harmonic. As shown in Figure 11, the peak values of total SDC of the three degradations have negligible variations and still maintain a linear relation with TOECs. There is almost no resolution loss by using the third harmonics. This indicates that the SDCs are relative values, which can be used for reconstruction to reduce the interference. Although the weakly third harmonic used is the noise effect generated when the second harmonic is actuated, it can also be applied to the reconstruction of material degradation in an ideal state. It can be concluded that the nonlinear harmonics of different orders are sensitive to the early-life material degradation. Moreover, this confirms that the NGW method can indeed predict the degree of degradation in the material.

### 5.6. Effect of the Separation Distance between Sensors

When the separation distance between two adjacent sensors is less than half of the wavelength of the primary mode, the reconstructions have the optimal resolution, as mentioned in Section 4.2. Therefore, it is meaningful to analyze the influence of separation distance of sensors on the resolution. Such issues have been discussed by Simonetti et al. [39] and Bernard et al. [42].

The situation which does not satisfy Equation (14) is discussed in this section, where the number is lower than 64. For simplicity, 32 sensors were selected from the 64 sensors in the array to form a small array, and 16 sensors were selected from the 32 sensors to form an even smaller array. The second harmonics were used for the reconstructions of two small arrays, and the probability distributions are shown in Figure 12. According to Equation (14), the array is composed of 32 sensors, D=78 mm, which belongs to the range $\frac{\lambda_{p}}{2}<D<\lambda_{p}$. In this case, the three degradations in the plate can still be relatively accurately located in the reconstructions. Given that $d_{N1}=d_{N2}=d_{N3}=60\;\mathrm{mm}$ and the smallest feature that can be reasonably accurately reconstructed through the estimation as $\sqrt{D\lambda_{p}}\approx 83.51\;\mathrm{mm}$ from the width of the Fresnel zone [20,21]. This indicates that the limited resolution has resulted in a poor reconstruction and a blurry edge of degradation. In the array composed of 16 sensors, D=156 mm—it belongs to the range $D>\lambda_{p}$. Here, the waves propagate as straight rays and their lobes may be unable to cover the entire detection area in the array network, and blind spots and constitute gaps are obvious. Consequently, it can be easily inferred that it is impossible to reconstruct the outlines of degradations, but simple positioning can still be achieved.

Therefore, when the separation distance between sensors is not enough to satisfy $D<\frac{\lambda_{p}}{2}$, the resolution is compromised, but the total numerical and computational cost can be reduced, e.g., in this work, the total WALLCLOCK TIME of 64 sensors is 25.6 h, that of 32 sensors is reduced to 1/2, and that of 16 sensors is reduced to 1/4; the reconstruction time of 64 sensors is 610 s, that of 32 sensors is reduced to 1/4, and that of 16 sensors is reduced to 1/16. In addition, it can also be inferred that when the number of sensors is not enough to support a full array, a limited number of sensors can be used, which is still useful for simple positioning of the defects.

### 5.7. Effect of Defect Types

It has been proved that nonlinear harmonics such as second and third harmonics are sensitive to the degradation of various degrees. Therefore, it is of interest to study whether the NGW method also has high sensitivity for detecting linear macroscopic defects. In order to highlight the effect of reconstruction, an actual aluminum (6061T6) plate was used for laboratory experiments of a size consistent with the simulations.

Verasonics UTA 160DH system was adopted as a testing instrument, which provides 128 independent transmitter channels and 128 independent receiver channels. This system contains the Vantage hardware unit, the Vantage software, and the “Host Controller”, a computer on which Matlab and the software have been installed. The hardware unit consists of the system backplane (BKP), transmit power controller (TPC), and a UTA Baseboard or ScanHead Interface (SHI).

Unlike the FE degradation with variation in TOECs, plasticine with absorbing guided wave effect is treated as the macroscopic defect in the experiments. As shown in Figure 13, a piece of plasticine with a diameter of 85 mm was placed at the (80, 50) coordinate pair of the aluminum plate. Taking the center of the aluminum plate as the origin point (0, 0), a circular array with a radius of 350 mm composed of 64 piezoelectric sensors (product model: PZT5h) was arranged. Since the experiment is no longer in an ideal environment, the sampling frequency and sampling points are adjusted by increasing the value. The experimental establishment of other aspects was the same as the methods used in Section 4.2. Figure 14 shows the time domain signals and the frequency spectra when the plasticine is on the vector of an *i–j* pair (the situation in this figure is i=10, j=45). In the experiment, there is an extra voltage and displacement conversion in the piezoelectric sensor, which is different from simulation. Therefore, the label of the vertical axis is the voltage. It is obvious that the signal amplitudes of the S0, A0 modes and the subsequent wave reflection are varying influenced by the macroscopic defect. The above indicates that larger SDCs will be generated.

The 60 kHz S0 Lamb mode was used as the primary mode, and the second harmonics were used in the NGW method. For experimental comparison, the NGW, attenuation, MP and covariance method were used for signal processing, respectively, and the probability distributions obtained are shown in Figure 15. Since plasticine is a viscoelastic material, the variation of surface waves caused by plasticine is more than that caused by degradation. As shown in Figure 15, the peak value of NGW method is 16.87, and that of covariance method is 9.391, which proves the secondary modes are more easily attenuated than the primary modes in the process of propagation. It is worth noting that since only one defect is on the material, there is basically no positioning deviation as the case of multiple degradations shows in Figure 8a. By detecting the SDCs of the peak prominences and peak values in the time domain signals, respectively, the obtained probability distributions of the attenuation and MP methods are plotted in Figure 15b,c. It can be observed that the position of plasticine is with a deviation. Besides, there are more serious artifacts in these two figures, which have a relatively large SDC resulting in reduced resolution. Overall, the covariance method more accurately reconstructed the position and size of the plasticine, which shows that the Lamb wave primary modes have exceptional sensitivity for detecting linear macroscopic defects. The nonlinear harmonic for NGW method is also sensitive to linear macroscopic defects. Although the resolution is only passable and the artifacts are large due to the low power flux, the defect location can still be detected.

## 6. Conclusions

In this work, we studied the nonlinear guided wave tomography which is used for detection early-life material degradation in metal plates. A generalized nonlinear acoustic modeling framework has been established which implements hyperelastic constitutive relations into Abaqus VUMAT subroutine. We developed a guided wave tomography signal processing platform with rich acoustic characteristics and compared the reconstruction performance of the NGW method with several other methods based on the results of detecting degradations and macroscopic defects. Numerical results validated that nonlinear guided wave tomography is capable of characterizing the severity of early-life material degradation. The NGW method has the potential capacity to simultaneously realize the qualitative detection of material degradation and the quantitative detection of the degradation degree. Furthermore, it was demonstrated that nonlinear tomography can still locate the degradation with a limited number of sensors. Finally, it can be deduced that the nonlinear harmonic is also sensitive to linear macroscopic defects.

However, when there are multiple defects on a material at the same time, the position of which in the probability distribution converges, leading to deviations. If the signal processing in the NGW method is enhanced, this problem may be improved. In addition, higher-order harmonics are extremely susceptible to the nonlinear noise of the systems, which can cause a decrease in the resolution. The nonlinear guided wave mixing has the potential quality to overcome this problem. By making full use of the multi-mode and dispersive characteristics of guided waves, it can effectively distinguish the nonlinear sources and improve the detection sensitivity. All the above-mentioned will be the next step in future research.

## Figures and Tables

**Figure 1 sensors-21-05498-f001:**
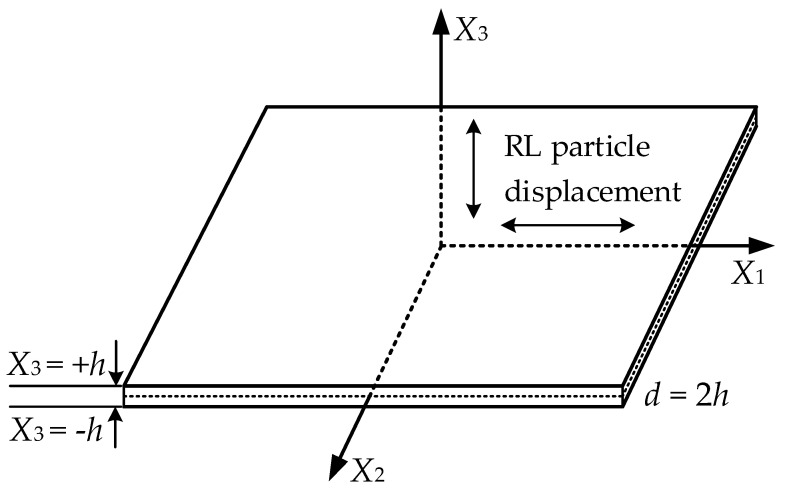
A free boundary plate in a reference coordinate system *X*_1_-*X*_2_-*X*_3_.

**Figure 2 sensors-21-05498-f002:**
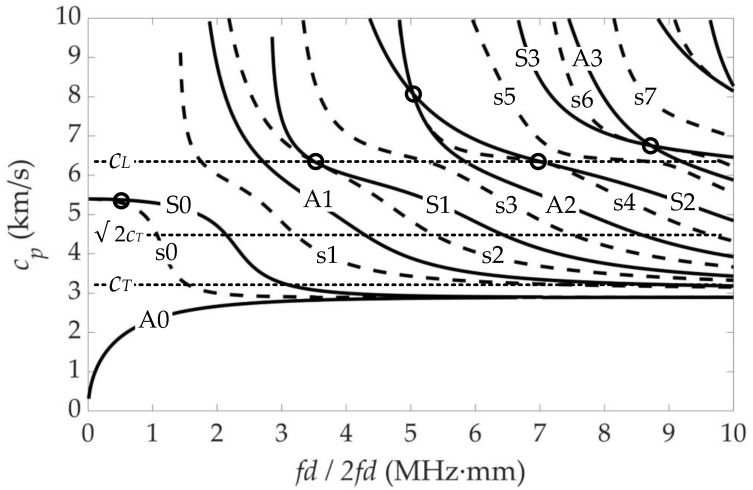
Internal resonance points of RL primary and secondary modes. The primary modes are shown as solid lines and plotted as *fd*. The secondary modes are shown as dash lines and plotted as 2 *fd*.

**Figure 3 sensors-21-05498-f003:**
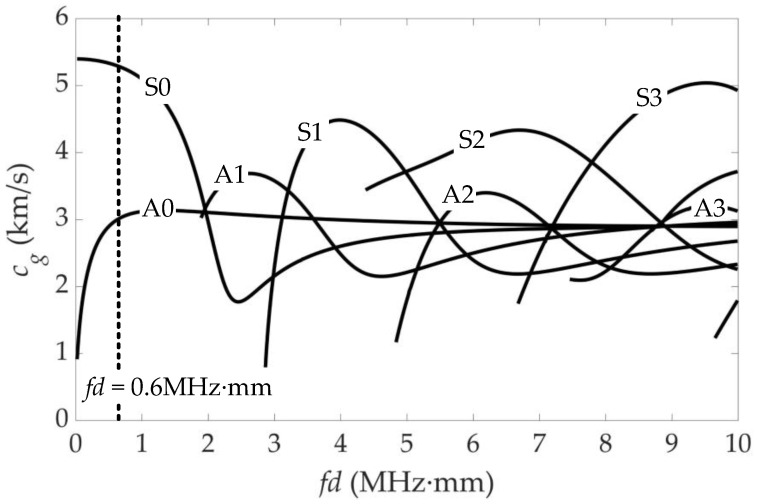
Group velocity dispersion curves of RL primary modes.

**Figure 4 sensors-21-05498-f004:**
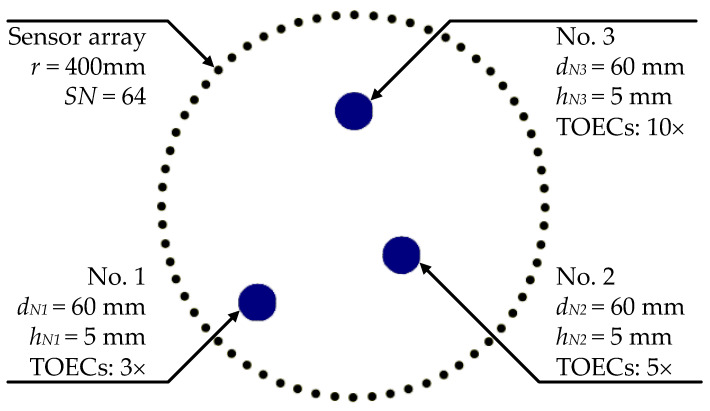
Aluminum plate model with early-life material degradations of different degrees. The other model is defined without these three degradations.

**Figure 5 sensors-21-05498-f005:**
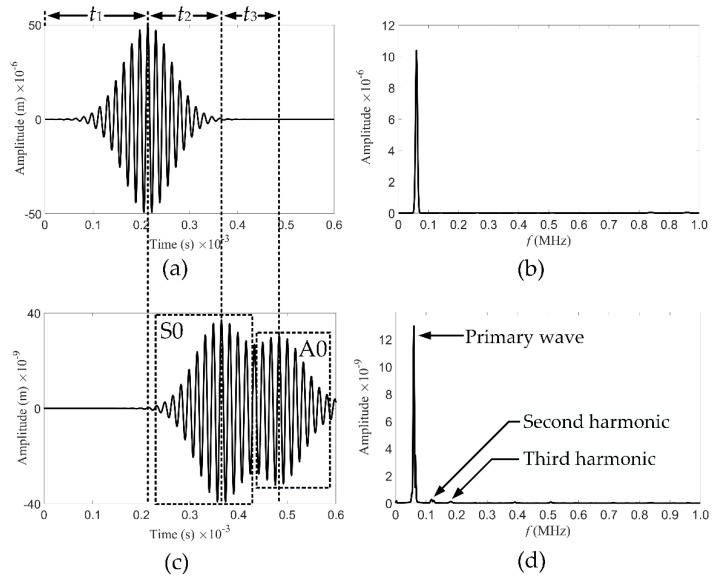
A sensor receives the signals actuated by itself and the signals actuated by the opposite sensor. (**a**) Time domain signals actuated by itself. (**b**) Frequency spectrum corresponding to (**a**). (**c**) Time domain signals actuated by the opposite sensor. (**d**) Frequency spectrum corresponding to (**c**). The central frequency of the excitation signals is 60 kHz.

**Figure 6 sensors-21-05498-f006:**
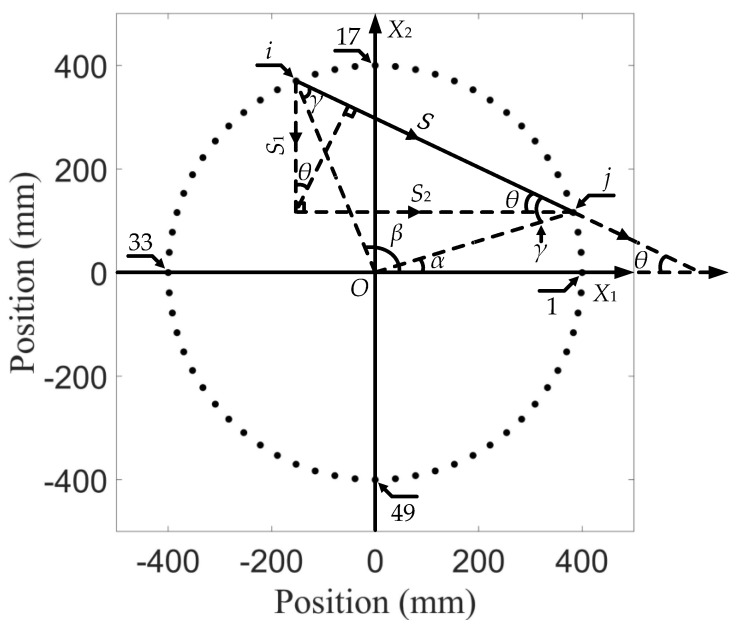
Trigonometric operations of the in-plane reception when the vector of an excitation–reception pair, *i–j*, is towards Quadrant IV of the coordinate system. Using the internal angle of the special triangle formed by the array origin and the two sensors, the *X*_1_ direction signal, *S*_1_, and the *X*_2_ direction signal, *S*_2_, are combined into the actuation–reception direction signal, *S*.

**Figure 7 sensors-21-05498-f007:**
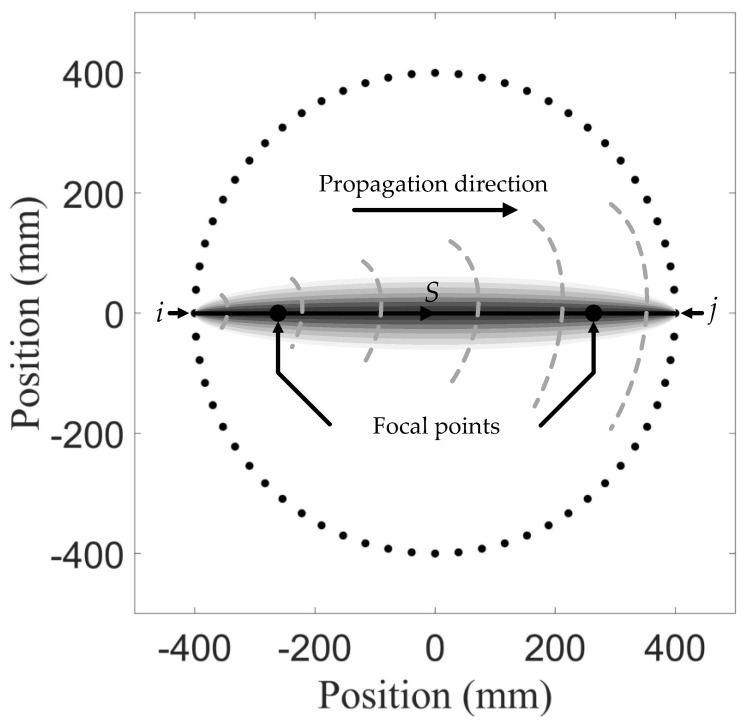
The ellipse of the elliptic algorithm, whose vertices of the ellipse are respectively regarded as the actuating signal sensor and the receiving signal sensor. The intensity on the line connecting the two vertices is the highest, indicating that the probability of degradation is the greatest. The probability is divided into 20 levels and decreases from the middle to both sides, and the probability of the edge and the periphery of the ellipse is zero.

**Figure 8 sensors-21-05498-f008:**
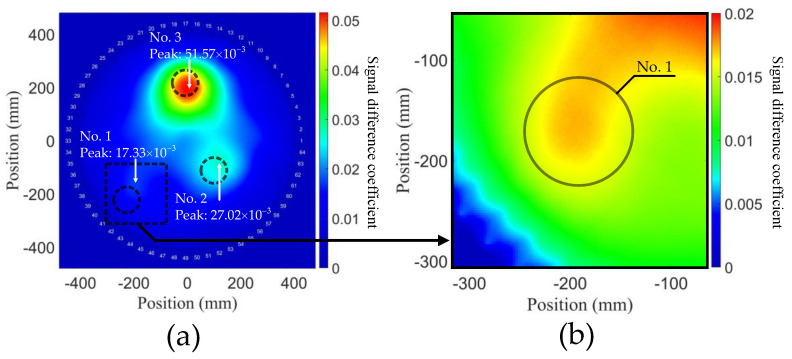
Probability distributions of aluminum plate model with early-life material degradations, using NGW method expressed with the second harmonics. (**a**) Reconstruction containing all degradations. The area covered by the circle corresponds to the positions of the three degradations in the FE model. (**b**) Zoom in on the square region around degradation No. 1 shown in (**a**).

**Figure 9 sensors-21-05498-f009:**
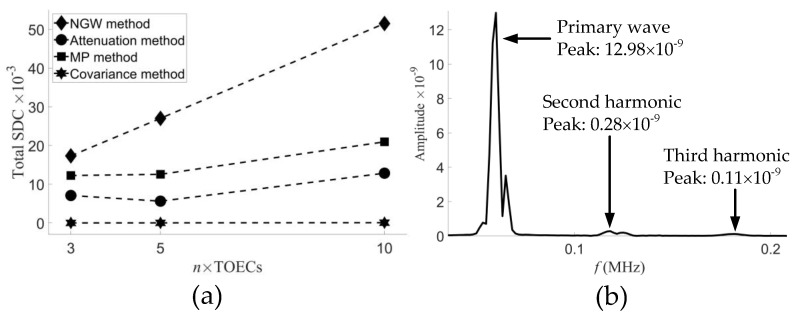
(**a**) The peak values of total SDC of the three degradations as a function of the TOECs. (**b**) The primary wave, second and third harmonic in the frequency spectrum of the signals received by the opposite sensor.

**Figure 10 sensors-21-05498-f010:**
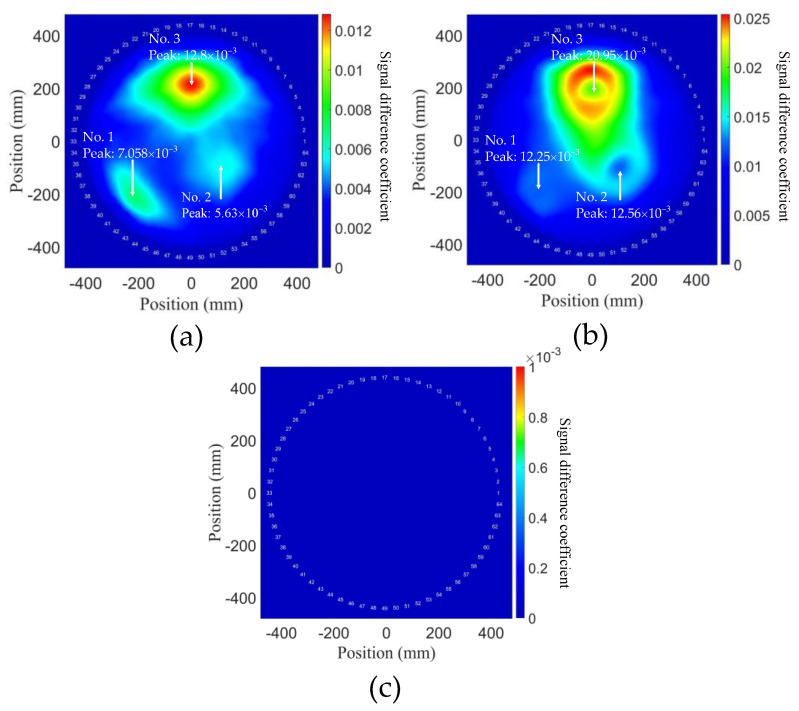
Probability distributions of aluminum plate model with early-life material degradations, using (**a**) attenuation method, (**b**) MP method and (**c**) covariance method.

**Figure 11 sensors-21-05498-f011:**
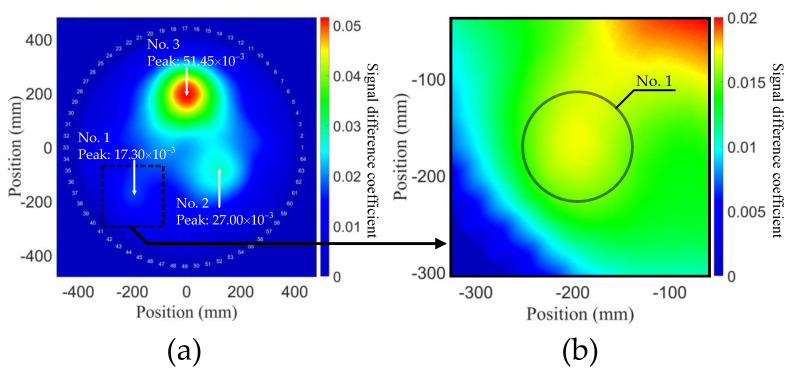
Probability distributions of aluminum plate model with early-life material degradations, using NGW method expressed with the third harmonics. (**a**) Reconstruction containing all degradations. (**b**) Zoom in on the square region around degradation No. 1 shown in (**a**).

**Figure 12 sensors-21-05498-f012:**
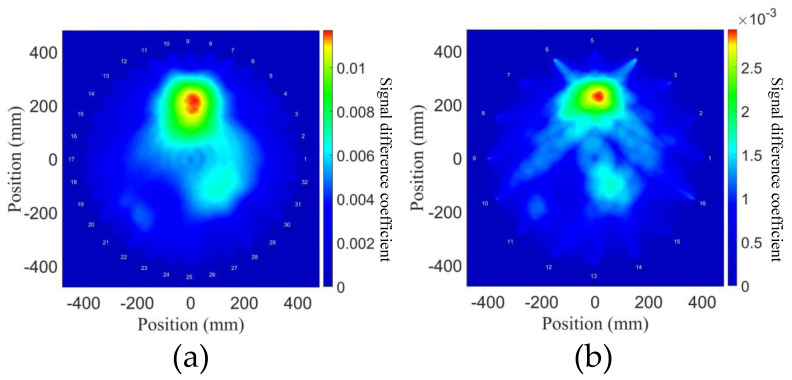
(**a**) Probability distributions for arrays composed of (**a**) 32 sensors and (**b**) 16 sensors, using the second harmonics for the NGW method.

**Figure 13 sensors-21-05498-f013:**
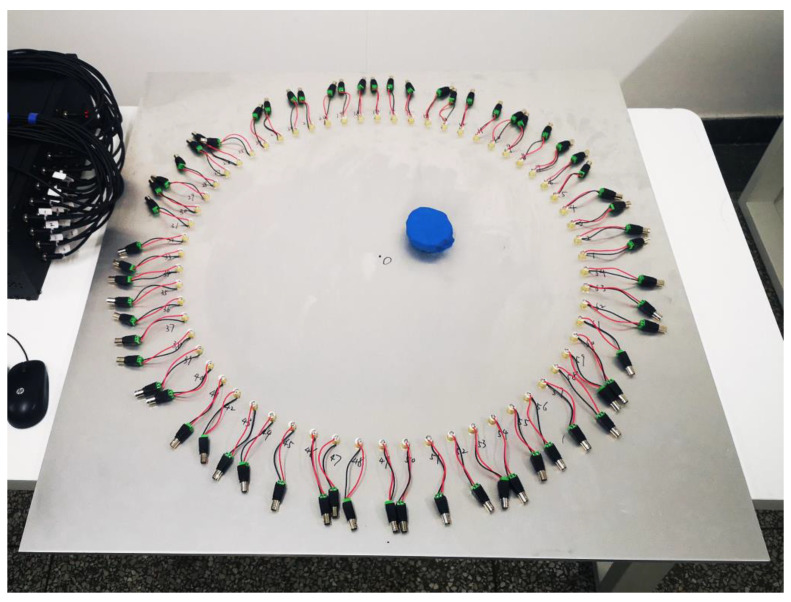
An aluminum plate with a piece of plasticine and 64 piezoelectric sensors.

**Figure 14 sensors-21-05498-f014:**
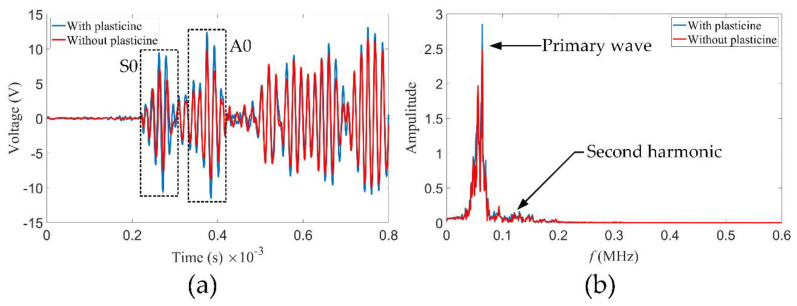
(**a**) Time domain signals when the plasticine is on the vector of an *i–j* pair. The signals of the aluminum plate without plasticine are shown as blue lines, and the ones of the aluminum plate with plasticine are shown as red lines. (**b**) Frequency spectrum corresponding to (**a**). The central frequency of the excitation signal is 60 kHz.

**Figure 15 sensors-21-05498-f015:**
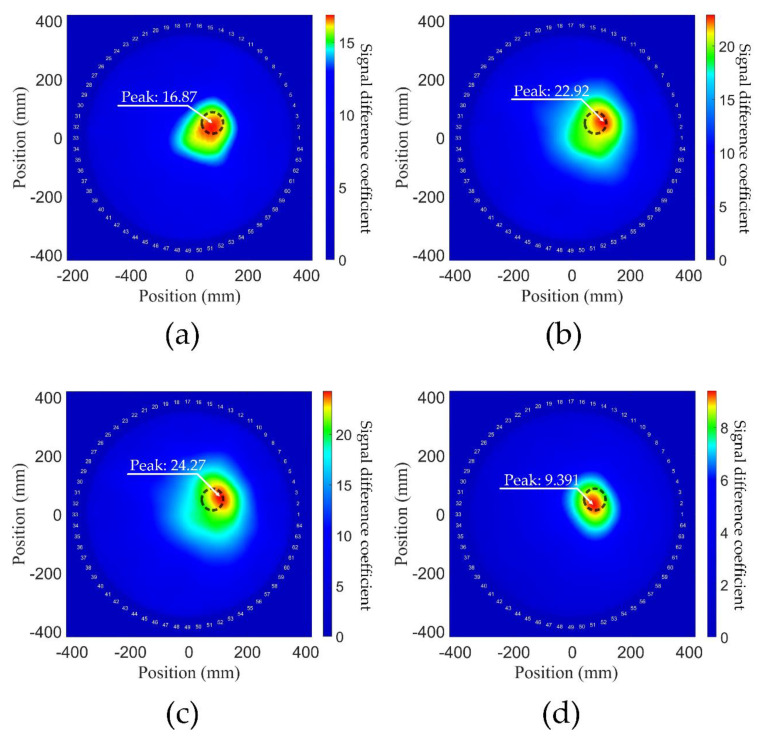
Probability distributions of the plasticine on the aluminum 6061-T6 plate in the laboratory reconstructed by (**a**) NGW method expressed with the second harmonics, (**b**) attenuation method, (**c**) MP method and (**d**) covariance method. The area covered by the circle corresponds to the positions of the plasticine in the experimental establishment.

## Data Availability

Not applicable.

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
