# Peer review of "Nonlinear Guided Wave Tomography for Detection and Evaluation of Early-Life Material Degradation in Plates"

_sensors, 2021, doi:10.3390/s21165498_

Round 1
Reviewer 1 Report
This paper developed a nonlinear guided wave tomography which is used for detection worked on evaluation of early-life material degradation in metal plates. Numerical results showed that nonlinear guided wave tomography is capable for characterizing the severity of early-life material degradation, but I think that the experiment could not validate the proposed method in this paper. Because the experiment scheme that used plasticine placed at the aluminum plate to simulate the material degradation is very different from the simulation model using Murnaghans TOECs to describe the material degradation. I suggest that corrosion damage can be considered, and the different degradation degree of the material can also be simulated. Moreover, the nonlinear guided wave experiment is very different from the ideal state of simulation actually, and for example, it may be meaningless to discuss the effect of harmonic orders due to the noise effect for extremely weak third harmonics. So it is very important to prove the effectiveness of the proposed method by the detection experiment of the material degradation.
Besides, there are other questions that need to answer:
1. Because the separation distance D between the sensors has to satisfy an ideal condition expressed by Eq.(14), does it mean that the higher frequency of the guided waves are used, the more sensors it needs? Is this method suitable for thin plate inspection?
2. Line 236-237 in P10, what is the X1 direction signal, S1, and the X2 direction signal, S2?
3. In section 5.7, the description of the experiment should be more detailed, for example, the picture of the experiment , the information on sensors, and the typical experimental signals.
Reviewer 2 Report
It is advisable to check the article for typos, e.g., line 34: Gol'dberg line 67: early-Life,...
Lines 75-84: references to sections in Roman numerals. In the text itself, references in the following also appear in Roman numerals, while the chapter titles themselves are numbered in Arabic.
I'm afraid that the formatting of references doesn't match the requirements. Please check and edit.
Introduction: Despite a sufficiently extensive introductory word, Horace Lamb has not mentioned.
The main contribution of the article is clearly defined in lines 66-74.
The units in the axis description of Figure 2 should be in MHz.mm.
I recommend a more detailed description of the meaning of nodes S0, S1, A0, A1, etc.…
I lack primary data about the FE model - e.g., number of nodes, number of elements, etc. Is it possible to reduce the model to one quarter to place the center of the coordinate system in the model's center or to one-eighth based on symmetry/antisymmetry?
Abaqus VUMAT subroutine - is it available? Could you please publish it as a supplementary file?
Is the term "lambda tr (E) I" in Equation 2 correct?
Where see the authors see their contribution in Chapters 2 and 3?
Line 283: the term "heat-map" would probably be more accurate than the term "heat."
(I assume that the size of the images will be solved by publishing the article online)
Reviewer 3 Report
Dear Authors,
Based on the reading of your work in my opinion you obtain a valuable paper in a topic related to evaluation of the plates degradation based on the nonlinear guided wave tomography. I recognise the efforts Authors have put into this work. However, in my opinion the manuscript needs to be improved (the minor revision is necessary).
Evaluation of the paper general remarks:
The structure of the article is clear and logical. The article contains a broadly presented state of the art in the field of theoretical analysis of propagation of elastic waves in solid media, mainly the use of Lamb waves. The Authors also point to the possibility of using the Murnaghan’s model to describe acoustic-elasticity and dynamic nonlinear elasticity in hyperelastic materials. At the end of chapter 1. Introduction Authors correctly determined the need for use of nonlinear guided wave tomography for detection and evaluation of early-life material degradation in metal plates. Modern techniques for assessing the condition of machine elements, including plates, which do not interfere with the material structure (the use of non-destructive testing methods) are desirable, especially in industry. All analyses included in chapters 2 to 5 contain information leading to the development of a model (platform) that allows visualization of early-life degradation in aluminum plates. However, what is the accuracy of the proposed model and the repeatability of the obtained results? In chapter 5.7, the results of laboratory tests of an aluminum sample of the same size as in the simulation are presented. Please add a photo of this sample during testing along with the test apparatus. Also, the article contains some editing errors, which are listed below.
Specific, mainly editorial remarks of the paper:
- Table I should be added to the main text of the manuscript, because it contains relevant material data.
- line 12, 66, 113, 226 ... - Please avoid writing pronouns like "We" in text, as this is highly unsuitable for academic and professional writing. Please use passive verb or different verb in active voice. Please check the entire text of the manuscript.
- why is a larger font used in certain parts of the text? i.e. line 189, 190, 192, 193, 194, 205, 207, 208, 210...
- in some places of the text, no punctuation mark ends before the text equations, i.e. line 240, 243. In my opinion, there should be a colon before the equation and at the end of the text.
- Please read the instructions on how to describe the references at the end of the article in the authors' guide and change it. Currently, the references at the end of the text are not in line with the journal requirements.
The article requires the above minor changes. I hope, these suggestions can help to improve the quality of this paper.
I wish You all the best.
Round 2
Reviewer 1 Report
Please add the frequency spectrum information for the signals Fig.13(b) like Fig.5(d) in Fig.13。
